# Infectious Keratitis: Characterization of Microbial Diversity through Species Richness and Shannon Diversity Index

**DOI:** 10.3390/biom14040389

**Published:** 2024-03-24

**Authors:** Domenico Schiano-Lomoriello, Irene Abicca, Laura Contento, Federico Gabrielli, Cinzia Alfonsi, Fabio Di Pietro, Filomena Tiziana Papa, Antonio Ballesteros-Sánchez, José-María Sánchez-González, Carlos Rocha-De-Lossada, Cosimo Mazzotta, Giuseppe Giannaccare, Chiara Bonzano, Davide Borroni

**Affiliations:** 1I.R.C.C.S.-G.B. Bietti Foundation, 00198 Rome, Italy; domenico.schiano@fondazionebietti.it (D.S.-L.); irene.albicca@fondazionebietti.it (I.A.); laura.contento@fondazionebietti.it (L.C.); 2Biolab SRL, Laboratorio di Genetica e Genomica Molecolare, Largo degli Aranci, 9, 63100 Ascoli Piceno, Italy; federico.gabrielli@laboratoriobiolab.it (F.G.); cinzia.alfonsi@laboratoriobiolab.it (C.A.); fabio.dipietro@laboratoriobiolab.it (F.D.P.); ngs@laboratoriobiolab.it (F.T.P.); 3Department of Physics of Condensed Matter, Optics Area, University of Seville, 41004 Seville, Spain; antbalsan@alum.us.es (A.B.-S.);; 4Department of Ophthalmology, Clínica Novovisión, 30008 Murcia, Spain; 5Regional University Hospital of Malaga, Hospital Civil Square, 29009 Malaga, Spain; carlosrochadelossada5@gmail.com; 6Department of Surgery, Ophthalmology Area, University of Seville, 41009 Seville, Spain; 7Siena Crosslinking Center, 53100 Siena, Italy; cgmazzotta@libero.it; 8Eye Clinic, Department of Surgical Sciences, University of Cagliari, 09121 Cagliari, Italy; giuseppe.giannaccare@gmail.com; 9DiNOGMI, University of Genoa and IRCCS San Martino Polyclinic Hospital, 16132 Genoa, Italy; oculistabonzano@gmail.com; 10Department of Ophthalmology, Riga Stradins University, LV-1007 Riga, Latvia; 11Eyemetagenomics Ltd., 71-75, Shelton Street, Covent Garden, London WC2H 9JQ, UK

**Keywords:** microbial keratitis, metagenomics, microbiome, *Proteobacteria*, microbial diversity

## Abstract

*Purpose*: To characterize microbial keratitis diversity utilizing species richness and Shannon Diversity Index. *Methods*: Corneal impression membrane was used to collect samples. All swabs were processed and analyzed by Biolab Laboratory (level V—SSN Excellence: ISO 9001:2015), Biolab Srl (Ascoli Piceno, Italy). DNA extraction, library preparation, and sequencing were performed in all samples. After sequencing, low-quality and polyclonal sequences were filtered out by the Ion software. At this point, we employed Kraken2 for microbial community analysis in keratitis samples. Nuclease-free water and all the reagents included in the experiment were used as a negative control. The primary outcome was the reduction in bacterial DNA (microbial load) at T1, expressed as a percentage of the baseline value (T0). Richness and Shannon alpha diversity metrics, along with Bray–Curtis beta diversity values, were calculated using the phyloseq package in R. Principal coordinate analysis was also conducted to interpret these metrics. *Results*: 19 samples were included in the study. The results exhibited a motley species richness, with the highest recorded value surpassing 800 species. Most of the samples displayed richness values ranging broadly from under 200 to around 600, indicating considerable variability in species count among the keratitis samples. *Conclusions:* A significant presence of both typical and atypical bacterial phyla in keratitis infections, underlining the complexity of the disease’s microbial etiology.

## 1. Introduction

Microbial keratitis is one of the most severe emergencies in ophthalmology and a leading cause of preventable blindness globally [1].

In more than 50% of cases, the pathogen responsible for a corneal infection is not identifiable. Etiological diagnosis is often challenging due to similar clinical presentations caused by a multitude of different pathogens. Additional confounding factors are the low specificity and sensitivity of currently available assays and difficulties in collecting suitable clinical specimens [2,3].

Conventional diagnostic methods (CDMs), including corneal scrapings for histopathological analysis, tissue culture, polymerase chain reaction (PCR) for detection of bacteria, fungal, and viral contaminants, and in vivo corneal confocal microscopy, are indeed successful in only 40 to 50% of cases. In addition, the achievement of a positive culture needs several days, whereas the rapid detection of pathogens is critical to establish a therapeutic strategy for microbial keratitis [4,5,6,7].

To overcome these difficulties, metagenomics relies on next-generation sequencing (NGS) techniques without microbial cultures or genome separation. There are two different approaches: the marker gene sequencing approach (or targeted amplicon sequencing) and the shotgun approach. The first one recognizes a microbial group in a sample using a specific gene obtained by PCR. The latter characterizes previously unknown microbes using random fragments of all DNA in a sample [8]. Both techniques allow the opportunity to focus on the ‘uncultured majority’ of prokaryotes, eukaryotes, and viruses that CDM cannot detect, including the previously unstudied 98% of existing microbes [9,10]. Moreover, traditional methods can only detect pathogens that can already be cultured or genetically identified, while NGS utilizes the hypothesis-free approach [11].

Metagenomics has already found a wide application in various fields of medicine. In 2008, the Human Microbiome Project (HMP) was launched is the US, aiming to define the microbial populations of gut, skin, mouth, nose, and urogenital tract [9], and the European Commission approved a project focused on the correlations between human intestinal microbiota and the number of disorders [12].

While skin, gut, the respiratory and urogenital tracts have been extensively described [13], the microbial diversity of the ocular surface remains poorly investigated. It is clear that the ability to identify previously unknown pathogens, which is the prerogative of metagenomics, is revolutionizing microbiology and different fields of medicine, including ophthalmology [14].

Metagenomics studies showed that the bacterial flora of the ocular surface in healthy patients appears different than previously characterized using culture techniques [15,16,17,18,19,20]. Studies comparing organisms identified using conventional cultures with 16S rDNA gene sequencing demonstrate that molecular techniques are able to identify more bacterial genera, including several uncultured bacteria [15,16]. A microbial community analysis of the conjunctiva revealed an average of 224 distinct bacterial phylotypes per individual (n = 4) [21]. Recently, the ocular surface microbiome of patients who underwent cataract surgery and used antimicrobial drops in the perioperative period or were affected by diabetes have been studied by metagenomics [22,23].

However, all these studies included eyes without infections.

As far as the application of this new technique in microbial keratitis is concerned, to date, it has not been sufficiently investigated in the literature. Omi et al. described a particular 16S rRNA sequencing analysis in four cases of corneal ulcer for the identification of pathogen [24].

Our group has already reported the application of shotgun sequencing in microbial keratitis with negative CDM [25].

The aim of our study is the evaluation of the microbial load in microbial keratitis to improve the understanding of infections.

## 2. Materials and Methods

### 2.1. Sample Collection

Inclusion criteria: corneal sample of patients with clinical suspected infection by bacteria, acanthamoeba, or fungi and at least a 1 mm corneal epithelial defect size.

Exclusion criteria: no sign of clinical infection and no epithelial defects.

A 5 mm Corneal Impression Membrane (CIM) (Millipore filter paper, with a pore size of 0.4 µm) was applied to the surface of the lesion for 5 s and placed in a tube with 1 mL of Amies medium, Copan ESwab^®^ (Copan Italia, Brescia, Italy). All swabs were processed and analyzed by Biolab Laboratory (level V—SSN Excellence: ISO 9001:2015), Biolab Srl (Ascoli Piceno, Italy).

### 2.2. DNA Extraction, Library Preparation, and Sequencing

Microbial DNA was isolated in a standardized manner with QIAamp DNA Microbiome Kit (Qiagen, Hilden, Germany) following the manufacturer’s protocol. Genomic DNA was fragmented with Ion Shear Plus Kit (Thermo Fisher, Waltham, MA, USA). A size of 350 bp was narrowly size selected by sample purification beads (Beckman Coulter Agencourt, Thermo Fisher, Waltham, MA, USA). Quality fragments were checked and quantified with Agilent TapeStation 4150 using Genomic DNA ScreenTape (Agilent Technologies, Santa Clara, CA, USA).

Probes containing sequencing adapters were hybridized to each DNA fragment. Libraries were tested for purity using Agilent High Sensitivity D1000 ScreenTape (Agilent Technologies, USA), and they were pooled together in equimolar proportions at the final concentration of 60 pM. Template preparations were performed with Ion Chef according to Ion 540 Kit-Chef protocol. Sequencing of the amplicon libraries was carried out on a 540 chip using the Ion Torrent S5 system (Thermo Fisher, Waltham, MA, USA). The resulting nucleotide-level data were analyzed to generate an accurate reference genome, compare variations within and between species, and differentiate between organisms.

After sequencing, low-quality and polyclonal sequences were filtered out by the Ion software. At this point, we employed Kraken2 for microbial community analysis in keratitis samples. Kraken2, a rapid taxonomic classification tool, assigns sequences to taxa by matching k-mers against a comprehensive genomic database. This method enables the accurate identification and quantification of microbial species, bypassing the need for traditional Z-score calculations and providing a direct, robust representation of the microbial diversity present in the ocular samples.

Nuclease-free water and all the reagents included in the experiment were used as a negative control.

### 2.3. Outcomes

The primary outcome was the reduction in bacterial DNA (microbial load) at T1, expressed as a percentage of the baseline value (T0).

Secondary outcomes were the microbial load at T1 in index and control eyes, patients’ adherence to therapy (questionnaire), discomfort (questionnaire), and pain after drops’ administration (questionnaire). In addition, we evaluated the influence of baseline variables (ocular comorbidities, concomitant use of ocular topical therapies, history of ocular surgeries) on the primary and secondary outcomes.

### 2.4. Statistical Analysis

The raw sequencing reads were processed using Kraken2 with PlusPFP database for precise taxonomic classification. On average, each sample yielded 3 million reads. Following the exclusion of host genome sequences, approximately 1.2 milion reads per sample were available for microbial analysis. Of these, 90% were attributed to bacteria, reflecting the predominance of bacterial pathogens in infectious keratitis. Fungal sequences were also detected, albeit in a smaller proportion, while viral sequences were not found to be significant. Richness and Shannon alpha diversity metrics, along with Bray–Curtis beta diversity values, were calculated using the phyloseq package in R. This computational analysis was further enhanced by principal coordinate analysis to interpret these diversity metrics comprehensively. All statistical tests applied were two-sided, with a significance threshold set at 5%. These analyses collectively provide a detailed understanding of the microbial diversity present in keratitis infections and highlight the substantial bacterial contribution relative to other microorganisms.

## 3. Results

### 3.1. Nineteen Samples Were Included in the Study

Species Richness and the Shannon Diversity Index were employed to assess microbial communities’ diversity. The results exhibited a motley species richness, with the highest recorded value surpassing 800 species. Most of the samples displayed richness values ranging broadly from under 200 to around 600, indicating considerable variability in species count among the keratitis samples (Figure 1A).

In Shannon Diversity Index analysis, the values fluctuate significantly, with one sample showing a peak diversity index near one, suggestive of both a high number of species and a balanced distribution of species abundances. Conversely, several samples had diversity indices below 0.2, pointing to less diverse microbial populations (Figure 1B).

### 3.2. Microbial Diversity in Keratitis Infections

The microbial communities’ composition analysis identified the *Proteobacteria* as the most abundant phyla constituting 51.6% of the microbial taxa present in the population.

The second most represented group is *Actinobacteria (Actinomycetota)* that comprises 21.4% of the community. *Firmicutes (Bacillota)* accounted for 10.3% of the microbiota, and *Cyanobacteria (Cyanobacteriota)* and *Bacteroidetes (Bacteroidota)* each constituted 9.4% of the total microbial composition. Twenty-six more phyla were identified, and each represented less than 1% of the population (Figure 2).

### 3.3. Dominant Microbial Species in Keratitis Samples

The study quantitatively analyzed the prevalence of microbial species within the samples, focusing on bacteria, and provided a detailed overview of their distribution and frequency of occurrence. The horizontal bar chart illustrates the read counts for each species, shedding light on their relative abundance in the sample pool. Notably, *Phocaeicola vulgatus* emerged as the most abundant bacterial species, followed by *Bifidobacterium adolescentis* and *Bacteroides uniformis*, indicating their significant presence (Figure 3).

*Faecalibacterium duncaniae* and *Bacteroides eggerthii* also showed substantial read counts, suggesting their consistent identification across the samples. *Faecalibacterium prausnitzii*, *Coprococcus* sp. *ART55/1*, and *Escherichia coli* were identified with moderate abundance, highlighting their role in the microbial composition of the samples.

Less prevalent but still noteworthy were *Parabacteroides distasonis*, reflecting a lower but still significant presence in the dataset. These findings underline the diversity of the bacterial landscape within the samples, with *Phocaeicola vulgatus* leading in prevalence, followed by other bacteria that contribute to the complex microbial environment.

## 4. Discussion

The current study presents an extensive characterization of microbial diversity within keratitis samples, utilizing species richness and the Shannon Diversity Index as primary metrics. The variability observed in species richness across samples underscores the polymicrobial nature of keratitis, which aligns with the findings of previous studies, suggesting that multiple microbial agents can contribute to the infection’s etiology.

The Shannon Diversity Index, which combines elements of richness and evenness, varied widely among our samples. In our analysis the values oscillate significantly, moving between a high Shannon index in some samples indicating not only a large number of species but also a more equitable distribution among them and samples with less diverse microbial populations. This could suggest a more stable and mature microbial community, which has been linked to increased resistance to pathogen invasion and colonization. Conversely, low diversity index values might reflect a community dominated by one or a few species, which could either indicate a recent perturbation or a community in the early stages of succession.

Interestingly, a high diversity index in keratitis may not necessarily correlate with a better prognosis. Instead, it may reflect a complex interplay of microbial interactions that could complicate treatment due to multiple resistance mechanisms and the potential for synergistic pathogenicity. On the other hand, lower diversity may suggest an infection driven by a single or limited number of pathogens, which could potentially be more amenable to targeted antimicrobial therapies. The relationship between high species richness and the severity or duration of keratitis symptoms remains to be elucidated. While a diverse microbiome is generally considered beneficial for host health, in the context of infection, a high level of microbial diversity may pose challenges for identifying and eradicating the causative agents.

In our study, the taxonomic distribution revealed a diverse microbial community dominated by several key phyla. The *Proteobacteria phylum*, the most abundant in our analysis, is known for its wide range of pathogenic species, which could be significant in the context of keratitis. *Actinobacteria* are recognized for their role in the normal flora of the human body and are also associated with various opportunistic infections. The analysis also identified the *Firmicutes phylum* that includes many common bacteria, some of which are known to be involved in eye infections. The presence of *Cyanobacteria* is noteworthy, given their rarity in clinical infections, suggesting a possible environmental influence on keratitis.

These findings indicate a significant presence of both typical and atypical bacterial phyla in keratitis infections, underlining the complexity of the disease’s microbial etiology. The predominance of *Proteobacteria* suggests its key role in the condition, warranting further investigation into its pathogenic mechanisms in the ocular environment. Further studies are necessary to elucidate the clinical implications of these results, which could lead to a better understanding of keratitis infections and potentially inform more effective treatment strategies.

Our study has multiple limitations, i.e., the sample size was modest, and the cross-sectional nature of the study precludes conclusions about the dynamics of microbial diversity over the course of infection. Longitudinal studies would be valuable to understand how microbial communities evolve during keratitis and in response to treatment. Additionally, further research should explore the functional implications of the identified species, as mere presence does not equate to pathogenicity or clinical significance.

## 5. Conclusions

In conclusion our findings contribute to a growing body of evidence that the microbial landscape of keratitis is complex and variable. This complexity highlights the necessity for personalized approaches to diagnosis and treatment. Understanding the microbial diversity in keratitis not only informs our grasp of the disease’s pathophysiology but also aids in the anticipation of therapeutic challenges, thereby guiding more effective management strategies.

## Figures and Tables

**Figure 1 biomolecules-14-00389-f001:**
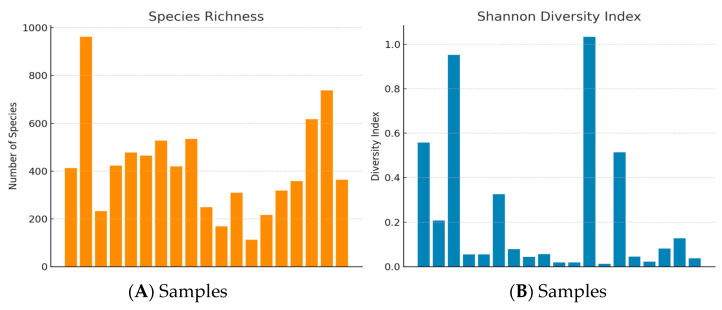
Species Richness and Shannon Diversity Index related to WGS analysis of keratitis samples. (**A**) Species richness analysis reveals between 200 and 600 different species in most keratitis samples. (**B**) Shannon Diversity Index ranges between several species with balanced distribution and less diversified population.

**Figure 2 biomolecules-14-00389-f002:**
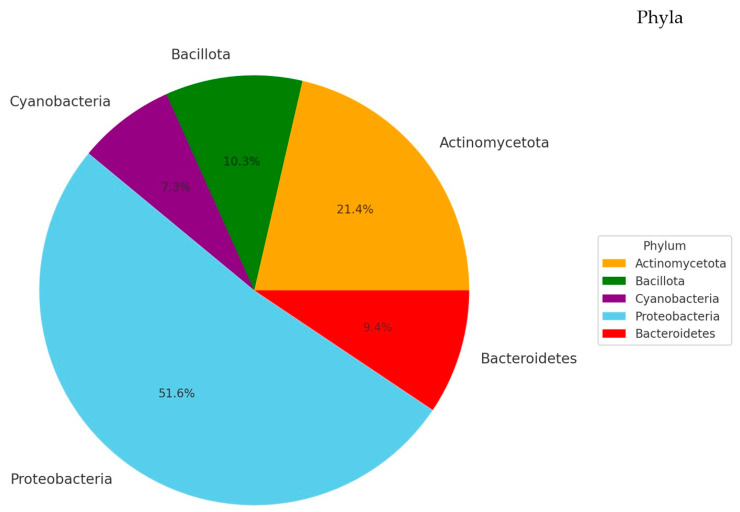
Microbial communities’ composition in keratitis samples. The most abundant phyla were *Proteobacteria* and *Actinobacteria* with 51.6% and 21.4%, respectively; *Cyanobacteria* represented 7.3%, and *Bacteroidetes* constituted 9.4%.

**Figure 3 biomolecules-14-00389-f003:**
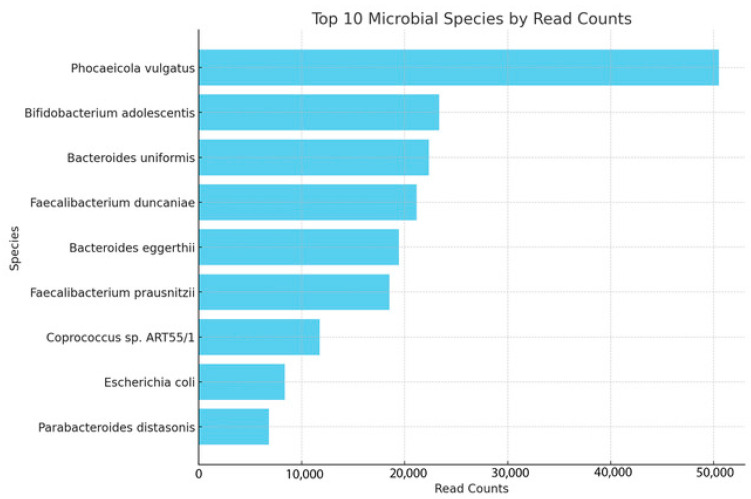
Top 10 most abundant species in keratitis samples. The horizontal bar reports the number of times each species was detected in the population. The most abundant species was *Phocaeicola vulgatus*.

## Data Availability

The datasets generated during and/or analyzed during the current study are available from the corresponding author on reasonable request.

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
