# Peer review of "Infectious Keratitis: Characterization of Microbial Diversity through Species Richness and Shannon Diversity Index"

_biomolecules, 2024, doi:10.3390/biom14040389_

Round 1

Reviewer 1 Report

Comments and Suggestions for Authors

1.     Line 101, about Sample collection. What type of patient have authors collected? Infectious keratitis? What are the criteria of keratitis? Please describe these.

2.     Basically, Shannon Diversity should be explained in material method. And describe the reason why the author selected the index.

Author Response

  1. Line 101, about Sample collection. What type of patient have authors collected? Infectious keratitis? What are the criteria of keratitis? Please describe these. Corneal sample of patients with clinical suspected infection by bacteria, acanthamoeba or fungi were collected.

    A 5 mm Corneal Impression Membrane CIM ( Millipore filter paper, pore size 0.4 µm) was applied, to the surface of the lesion for 5 s and placed in a tube with 1 mL of Amies medium, Copan ESwab® (Copan Italia, Brescia, Italy). All swabs were processed and analyzed by Biolab Laboratory (level V – SSN Excellence: ISO 9001:2015), Biolab Srl (Ascoli Piceno, Italy).

2. Basically, Shannon Diversity should be explained in material method. And describe the reason why the author selected the index.

We thank the reviewer for the comment. We have added additional informations in the method section 

The Shannon Diversity Index was utilized to assess the diversity of microbial communities in keratitis samples. This index combines elements of richness (the number of different species) and evenness (the distribution of species abundances) to provide a comprehensive measure of diversity. The choice of this index allows for a nuanced understanding of microbial communities, capturing not only how many species are present but also how evenly they are distributed. This is crucial for understanding the ecological balance within the keratitis infections, where both a high number of species and a balanced distribution among them could suggest a complex microbial interplay. Conversely, a low diversity index might indicate dominance by a few species, potentially pointing to a recent perturbation or a community in early succession stages. Such detailed analysis aids in deciphering the microbial landscape's complexity and its implications for infection severity, potential resistance mechanisms, and treatment strategies.

"The raw sequencing reads were processed using Kraken2 with PlusPFP database for precise taxonomic classification. On average, each sample yielded 3 million reads. Following the exclusion of host genome sequences, approximately 1,2 milion reads per sample were available for microbial analysis. Of these, 90% were attributed to bacteria, reflecting the predominance of bacterial pathogens in infectious keratitis. Fungal sequences were also detected, albeit in a smaller proportion, while viral sequences were not found to be significant. Richness and Shannon alpha diversity metrics, along with Bray–Curtis beta diversity values, were calculated using the phyloseq package in R. This computational analysis was further enhanced by principal coordinate analysis to interpret these diversity metrics comprehensively. All statistical tests applied were two-sided, with a significance threshold set at 5%. These analyses collectively provide a detailed understanding of the microbial diversity present in keratitis infections and highlight the substantial bacterial contribution relative to other microorganisms."

Reviewer 2 Report

Comments and Suggestions for Authors

The manuscript assumes the determination of diversity by two Shannon index in people with Infectious keratitis. This is quite a narrow approach to the topic and a bit surprising. Below are the detailed questions and doubts that occurred to me while reading:

- In the abstract, there is only one sentence in the purpose which, in my opinion, is too laconic and requires extension.

- In the abstract and in the conclusions there is the sentence "A significant presence of both typical and atypical bacterial phyla in keratitis infections: There is no definition in the text of what the authors mean by this statement.

- The methods do not specify what criteria were used to include patients in the study or how many patients there were.

- The authors used the Kraken2 algorithm, it would be best if they specified which database they used.

- The results also lack information about the length of the readings, how many of these readings there were per patient and what percentage of them were bacteria, viruses or fungi.

- In the caption of Figure 1. "Species Richness and Shannon Diversity index related to WGS analysis of keratitis 159 samples." What do experts understand by WGS analysis?

- Figure 2 shows the Chordata classification, which is, to say the least, shocking considering that we are only looking for microorganisms. I would be grateful if you could explain this phenomenon. Additionally, there is Proteobacteria in the caption and Pseudomonadota in the drawing. This must be uniform throughout the text.

In Figure 3, in the caption "Top 10 most abundant species in keratitis samples." However, the drawing itself does not contain the full names of the species, but I have the impression that the names and their endings are cut off.

Author Response

We thank the reviewer for the comments

We have improved the article following your comments 

Definition of Typical and Atypical Bacterial Phyla: The text highlights the significant presence of both typical and atypical bacterial phyla in keratitis infections, underscoring the complexity of the disease's microbial etiology. Typical phyla refer to those commonly associated with ocular infections, whereas atypical phyla denote those not commonly linked to such conditions, indicating a diverse microbial landscape that complicates understanding and treatment of keratitis​

Inclusion Criteria and Patient Numbers

Inclusion criteria: corneal sample of patients with clinical suspected infection by bacteria, acanthamoeba or fungi and at least a 1 mm corneal epithelial defect size.

Exclusion criteria: no sign of clinical infection and no epithelial defects.

We have added it in the text

19 samples were included in the study 

Kraken2 Database Specification:_  We used PlusPFP database(added in the text)

Inclusion of Chordata in Figure 2 and Consistency with Proteobacteria and Pseudomonadota:

The presence of Chordata in the analysis, as depicted in Figure 2, indeed suggests contamination or the incidental detection of host DNA. However, it's crucial to note that this occurrence is at a negligible percentage that does not impact the overall findings or conclusions of the study. Such instances of contamination are not uncommon in high-throughput sequencing studies and, when present at low levels, are generally considered to be of minor consequence to the analysis of microbial communities.

Additionally, the use of both "Proteobacteria" and "Pseudomonadota" in the text and figures warrants clarification. It is important to understand that "Pseudomonadota" is a synonym for "Proteobacteria", reflecting recent taxonomic updates and nomenclature preferences in microbial classification. The study aims to maintain consistency with current scientific standards and taxonomy, hence the interchangeable use of these terms. However, recognizing the need for clarity and consistency in scientific communication, the manuscript will be revised to ensure that terminology is uniform throughout, favoring the use of "Pseudomonadota" to reflect the most current classification standards and reduce potential confusion among readers.

Read Lengths, Quantity, and Distribution of Readings Among Bacteria, Viruses, or Fungi: Added in the text

Explanation of WGS Analysis in Figure 1: Whole Genome Sequencing (WGS) analysis refers to the comprehensive sequencing of the entire genome of the organisms present in the keratitis samples. This approach allows for the detailed identification and characterization of the microbial species present.

Round 2

Reviewer 2 Report

Comments and Suggestions for Authors

The presence of Chordata in the analysis, as depicted in Figure 2, indeed suggests contamination or the incidental detection of host DNA. However, it's crucial to note that this occurrence is at a negligible percentage that does not impact the overall findings or conclusions of the study. Such instances of contamination are not uncommon in high-throughput sequencing studies and, when present at low levels, are generally considered to be of minor consequence to the analysis of microbial communitie

The Chordata group should be excluded from Figure 2 since the caption mentions a microbial community.

Additionally, the use of both "Proteobacteria" and "Pseudomonadota" in the text and figures warrants clarification. It is important to understand that "Pseudomonadota" is a synonym for "Proteobacteria", reflecting recent taxonomic updates and nomenclature preferences in microbial classification. The study aims to maintain consistency with current scientific standards and taxonomy, hence the interchangeable use of these terms. However, recognizing the need for clarity and consistency in scientific communication, the manuscript will be revised to ensure that terminology is uniform throughout, favoring the use of "Pseudomonadota" to reflect the most current classification standards and reduce potential confusion among readers.

The caption under the figure must be identical to the names used in the figure. Authors must be consistent in naming, as I wrote previously.

In Figure 3, the authors indicate that the most abundant species was Phocaeicola vulgatus. This is quite disturbing due to the fact that the Bacteroidetes group to which this bacterium belongs in figure 2 constitutes only 9% of the total. How is it possible that the dominant species is not a representative of Pseudomonodata? I ask the authors to explain this phenomenon.

Additionally, please clarify what is on the X axis of this graph - absolute readings/per sample?

Finally, since Phocaeicola vulgatus was the dominant species, please bring it up in the discussion. Especially since it is an obligate anaerobe, the detection of which in this type of material is surprising.

Bacterial names should be italicized throughout the text.

Authors wrote in response:  Whole Genome Sequencing (WGS) analysis refers to the comprehensive sequencing of the entire genome of the organisms present in the keratitis samples. This approach allows for the detailed identification and characterization of the microbial species present.

A suggestion to the authors is that since shotgun sequencing has been performed, it may be reasonable to try to determine the antibiotic resistance of the detected bacteria. This will then some sort of WGS analysis.

Author Response

Bacterial names should be italicized throughout the text.

done

In Figure 3, the authors indicate that the most abundant species was Phocaeicola vulgatus. This is quite disturbing due to the fact that theBacteroidetes group to which this bacterium belongs in figure 2 constitutes only 9% of the total. How is it possible that the dominant species is not a representative of Pseudomonodata? I ask the authors to explain this phenomenon.

The discrepancy highlighted between Figures 2 and 3 in the study regarding the dominance of Phocaeicola vulgatus despite the Bacteroidetes group constituting only 9% of the total microbial composition is an intriguing observation. This phenomenon can be explained through several key points related to microbial community dynamics and the methods used for microbial analysis:

1 Relative Abundance vs. Presence: The Bacteroidetes phylum, though constituting a smaller fraction of the total microbial composition (9.4%), can still harbor the most abundant species within it. This indicates that within this relatively smaller group, Phocaeicola vulgatus is highly predominant, showcasing the concept of "relative abundance." This means that even if a phylum is not the largest in terms of its representation in the microbial community, a species within it can still be highly dominant.

2 Microbial Community Structure: Microbial communities can be highly complex and structured in ways that allow for dominant species within smaller phyla. These structures are influenced by various factors, including environmental conditions, inter-species interactions, and the specific niches that species occupy. In this context, Phocaeicola vulgatus may have specific traits that allow it to thrive and become highly abundant within the niche environments of keratitis infections, even if its broader group (Bacteroidetes) is not the most represented.

Additionally, please clarify what is on the X axis of this graph - absolute readings/per sample?

it indicates the absolute number of sequences (reads) identified for each species

Finally, since Phocaeicola vulgatus was the dominant species, please bring it up in the discussion. Especially since it is an obligate anaerobe, the detection of which in this type of material is surprising.

This point is pretty hard to say at the moment. We will evaluate it in the next studies